# Generating long-distance magnetic couplings with flat bands

G. Bouzerar* and M. Thumin

*Université Grenoble Alpes, CNRS, Institut NEEL, F-38042 Grenoble, France*

(Dated: February 28, 2025)

One of the great challenges for the large-scale development of quantum technologies is to generate and control the entanglement of quantum bits through interactions of sufficiently long range. Two decades ago, spin chains have been proposed for quantum communication. Unfortunately, couplings are of very short range in general which drastically limits the communication to very short distances. Here, we demonstrate that the presence of flat bands (FBs) can trigger very long-distance magnetic couplings in spin chains. Furthermore, we show that the typical decaying lengthscale is directly related to the quantum metric of the flat band eigenstates. We believe that our unexpected findings could open up an alternative route to enable long-distance quantum communication.

PACS numbers: 75.50.Pp, 75.10.-b, 75.30.-m

The last two decades have seen the emergence of the quantum era, which has become a flagship of modern physics due to its enormous potential for technological applications such as quantum computers, metrology, quantum sensing, communication, cryptography and cybersecurity [1–6]. Long-distance quantum communication requires lossless transfer of a quantum state (qubit) between distant quantum registers and processors. Since direct interactions between qubits decrease extremely rapidly with distance, efforts are being made to explore ways of achieving entanglement between qubits indirectly. Two decades ago, S. Bose has suggested that quantum spin chains could be an efficient channel for short distance quantum communication [7, 8]. Since these pioneering works, the study of transmission of data in linear quantum registers has been the subject of many investigations [9–12]. Experimentally, evidence of long-distance entanglement in spin chains was recently presented in Refs. [13, 14]. In particular, in Ref.[13], it has been revealed that unpaired $S = 1/2$ spins can exhibit effective couplings of significant amplitude ($\approx 3$ $K$) at distances as large as 20 $nm$. Identifying the mechanisms that can significantly increase the effective coupling between qubits far apart is one of the major challenges of quantum technology. In this study, we demonstrate that flat-band (FB) materials could be used to amplify the magnetic couplings between widely separated spins. The mechanism that enables long-range entanglement is connected to a geometric characteristic of FB eigenstates known as the quantum metric (QM). The QM is the real part of the quantum geometric tensor, and provides a measure of the square of the typical spread of FB eigenstates [15, 16]. Physics in FBs has recently emerged as a new field in the physics of strongly correlated electronic systems. [17–20].

To address this issue, we consider two different types of one-dimensional lattices: the stub lattice (Fig. 1(a)) and the diamond chain (Fig. 1(c)). Both exhibit a FB located at $E = 0$. The former offers a great advantage: the tuning of the (AB) hopping allows that of the gap

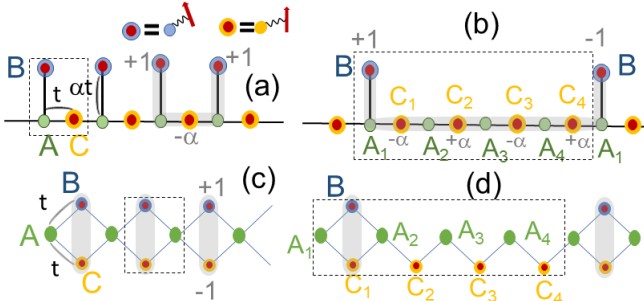

Figure 1. Sb[1] and Sb[4] are depicted in (a) and (b), Dd[1] and Dd[4] in (c) and (d). $B$ and $C$-orbitals are coupled to localized spins (red dots), dashed boxes depict the unit cell. The CLS and weight on $B/C$-orbitals are highlighted by grey regions and fonts. The hoppings are described in the text.

and of the QM. For convenience, we define $\mathcal{A}$, $\mathcal{B}$ and $\mathcal{C}$ as the sublattices of the $A$, $B$ and $C$-orbitals. Dilute versions of these chains, called Sb[n] (respectively Dd[n]), are obtained by removing $n-1$ successive orbitals in $\mathcal{B}$ of the stub (respectively diamond) chain. As their parent counterpart, they are bipartite and exhibit a FB at $E = 0$. Sb[n] and Dd[n] contain $2n+1$ orbitals/cell, the size of the unit cell being $a_n = na$. As an illustration, Sb[4] and Dd[4] are depicted in Fig. 1(b) and Fig. 1(d). The magnetic chains considered here are composed of both non-magnetic atoms located in sublattice $\mathcal{A}$ and magnetic ones occupying the $\mathcal{B}$ and $\mathcal{C}$. We believe that such a geometry could be designed experimentally with metal-organic frameworks (MOFs) [21–24]. By combining organic and inorganic building blocks, MOFs provide a chemically versatile and flexible platform for the development of new families of magnetic materials. MOFs offer numerous advantages, such as the ability to combine organic linkers at will to enable the synthesis of tunable structures with controllable physical properties.

For both Sb[n] and Dd[n] the Hamiltonian reads,

$$\widehat{H} = \sum_{\langle i\lambda, j\lambda' \rangle, \sigma} t_{ij}^{\lambda\lambda'} \hat{c}_{i\lambda\sigma}^{\dagger} \hat{c}_{j\lambda'\sigma} + J \sum_{i\lambda \in \mathcal{B}, \mathcal{C}} \widehat{\mathbf{s}}_{i\lambda} \cdot \mathbf{S}_{i\lambda}. \quad (1)$$

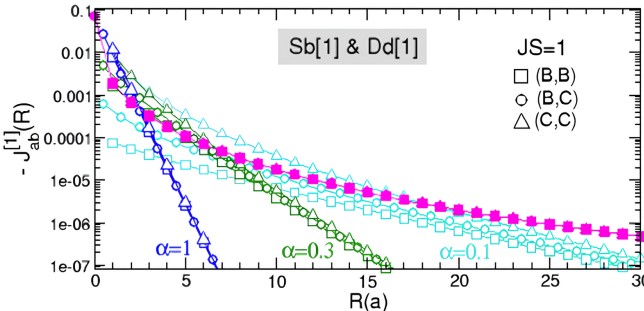

Figure 2. $-J_{ab}^{[1]}(R)$(in units of $t$) in Sb[1] and Dd[1] as a function of the distance $R$ for $|JS| = t$, where $(a, b) = (B, B)$, $(B, C)$ and $(C, C)$. For the Sb[1] chain we have considered $\alpha = 0.1$, 0.3 and 1. The couplings in the Dd[1] chain correspond to the filled magenta symbols.

$\hat{c}_{i\lambda\sigma}^{\dagger}$ creates an electron with spin $\sigma = \uparrow, \downarrow$ in the orbital $\lambda$ of the $i$-th cell. $\langle i\lambda, j\lambda' \rangle$ means nearest-neighbours. In Sb[n] chains (see Fig.1(a)) $t_{ij}^{\lambda\lambda'} = -t$ along the chain and $t_{ij}^{\lambda\lambda'} = -\alpha t$ in the perpendicular direction. In Dd[n] chains $t_{ij}^{\lambda\lambda'} = -t$ along and in the out-of-chain direction. $J$ is the local coupling between the localized classical spin $\mathbf{S}_{i\lambda} = S.\mathbf{e}_{i\lambda}$ at $\mathbf{r}_{i\lambda}$ ($\mathbf{e}_{i\lambda}$ being a unit vector) and that of the itinerant carrier, $\hat{s}_{i\lambda}^{a} = \hat{c}_{i\lambda\alpha}^{\dagger}[\hat{\sigma}^{a}]_{\alpha\beta}\hat{c}_{i\lambda\beta}$ where $a = x, y$, and $z$ and $\hat{\sigma}^a$ are the Pauli matrices. In what follows, we set $t = 1$ and $JS$ is expressed in units of $t$. Here, we consider half-filled systems.

First, we assume $JS = 0$ and recall the nature of the compact localized states (CLS) which are FB eigenstates that possess non-vanishing weights on a finite (minimal) number of orbitals [25]. In Dd[n] chains, for any $n$ the CLS is given by $|CLS\rangle_{Dd[n]} = \frac{1}{\sqrt{2}}(|B, i\rangle - |C, i\rangle)$, $i$ being the cell index which implies a vanishing QM. The situation differs in Sb[n] chains for which $|CLS\rangle_{Sb[n]} = \frac{1}{\sqrt{n\alpha^2+2}}(|B, i\rangle + (-1)^n|B, i+n\rangle + \alpha \sum_{i=1}^{n}(-1)^i|C, i\rangle)$[26]. Removing $B$-orbitals leads to a spreading of the CLS and hence to a strong increase of the QM as $n$ increases [27]. The CLSs are illustrated in Fig.1.

The coupling between two spins located at $r_{i\lambda}$ and $r_{j\lambda'}$ is given by [28],

$$J_{\lambda\lambda'}(R) = \frac{(JS)^2}{2}\int_{-\infty}^{+\infty}\chi_{ij}^{\lambda\lambda'}(\omega)f(\omega)d\omega, \qquad (2)$$

where $R = r_{i\lambda} - r_{j\lambda'}$ and the generalized susceptibility $\chi_{ij}^{\lambda\lambda'}(\omega) = -\frac{1}{\pi}\Im\left[G_{ij\uparrow}^{\lambda\lambda'}(\omega)G_{ji\downarrow}^{\lambda'\lambda}(\omega)\right]$. The Green's function $\hat{G}_{\sigma}(\omega) = (\omega + i\eta - \hat{H}_{\sigma})^{-1}$, $\hat{H}_{\sigma}$ being the Hamiltonian in the spin sector $\sigma$ ($\sigma = \uparrow, \downarrow$), $\eta$ mimics an infinitesimal inelastic scattering rate and $f(\omega) = \frac{1}{e^{(\omega-\mu)/k_BT} + 1}$ is the Fermi-Dirac distribution. Here, we consider half-filled systems ($\mu = 0$) at $T = 0\,K$.

Remark that $J_{\lambda\lambda'}(r) \geq 0$ (resp. $J_{\lambda\lambda'}(r) \leq 0$) means antiferromagnetic (resp. ferromagnetic) coupling. Eq. (2) is derived for classical spins and implies that

the corresponding effective Heisenberg Hamiltonian reads $\hat{\mathcal{H}}_{Heis} = \frac{1}{2}\sum_{i\lambda\neq j\lambda'}J_{\lambda\lambda'}(r_{ij}^{\lambda\lambda'})\mathbf{e}_{i\lambda}\cdot\mathbf{e}_{j\lambda'}$.

The calculation of the couplings requires the knowledge of the ground-state, hence that of the underlying localized spin texture. Because Sb[n] and Dd[n] chains are bipartite, the ground-state at $T = 0\,K$ is the ferromagnetic one where the localized spins of $\mathcal{B}$ and $\mathcal{C}$ sublattices are parallel to each other [29, 30].

For what follows, we define $E_k^{l\sigma}$ and $|\psi_k^{l\sigma}\rangle$ the eigenvalues and eigenvectors of $\hat{H}^{\sigma}$, where $\sigma = \uparrow, \downarrow$, $l$ is the band index, and $k$ the momentum. In Sb[n] and Dd[n] chains, the FBs are located at $E_{FB} = \pm\frac{JS}{2}$. The coupling between two spins located at $r_{i\lambda}$ and $r_{j\lambda'}$ can be decomposed into,

$$J_{\lambda\lambda'}(R) = \sum_{pq}I_{\lambda\lambda'}^{pq}(R), \qquad (3)$$

where $p$ (resp. $q$) is the band index in the $\uparrow$ (resp. $\downarrow$) sector and,

$$I_{\lambda\lambda'}^{pq}(R) = \frac{(JS)^2}{2N_c^2}\sum_{k,k'}e^{i(k-k')R}A_{k\lambda,k'\lambda'}^{pq}\frac{f(E_k^{p\uparrow}) - f(E_{k'}^{q\downarrow})}{E_k^{p\uparrow} - E_{k'}^{q\downarrow}}, (4)$$

$N_c$ is the number of cells, and $A_{k\lambda,k'\lambda'}^{pq} = \langle\lambda_{k,\uparrow}|\psi_k^{p\uparrow}\rangle\langle\psi_k^{p\uparrow}|\lambda'_{k,\uparrow}\rangle\langle\lambda'_{k',\downarrow}|\psi_{k'}^{q\downarrow}\rangle\langle\psi_{k'}^{q\downarrow}|\lambda_{k',\downarrow}\rangle$ where $|\lambda_{k,\sigma}\rangle = \hat{c}_{k\lambda\sigma}^{\dagger}|0\rangle$, $\hat{c}_{k\lambda\sigma}^{\dagger}$ being the Fourier transform of $\hat{c}_{i\lambda\sigma}^{\dagger}$. For convenience we define $J_{ab}^{[n]}(R)$ the magnetic couplings between spins, in the Dd[n] or Sb[n] chains, $a, b$ being orbital indices.

First, we discuss the couplings in the reference chains ($n = 1$). Figure 2 represents $-J_{ab}^{[1]}(R)$ as a function of $R$ in Sb[1] and Dd[1] chains for $|JS| = 1$ with $(a, b) = (B, B)$, $(B, C)$ and $(C, C)$. For the stub chain we have chosen 3 different values of $\alpha$ (0.1 , 0.3 and 1). In all cases, the couplings are negative confirming the ferromagnetic nature of the ground-state and the couplings decay exponentially ($J_{ab}^{[1]}(R) \simeq e^{-R/\xi}$). Furthermore, for $\alpha = 1$, the couplings for these three sets of pairs are almost identical. As $\alpha$ reduces, we observe at small distances strong deviations between the couplings, $|J_{CC}^{[1]}(R)| \gg |J_{BB}^{[1]}(R)|$, which could be anticipated since $B$-orbitals become weakly coupled to the chain. When $\alpha \gg JS$, our analysis reveals that the $(B, B)$ coupling reduces to the FB-FB contribution: $J_{BB}^{[1]}(R) \propto \alpha^2|JS|e^{-R/\xi}$ where $\xi = a/2\alpha$ ($\alpha$ being small enough). In the opposite regime ($JS \gg \alpha$), the $(B, B)$ coupling reduces to the contribution involving the filled dispersive band in the $\uparrow$-sector and the empty one in the $\downarrow$-sector: $J_{BB}^{[1]}(R) \propto -\alpha^{7/2}\frac{t^2}{|JS|}\frac{1}{\sqrt{R}}e^{-R/\xi}$ where $\xi = \sqrt{2}a/3\alpha$. These analytical expressions are derived in the supplementary material (SM) [31].

We now consider the case of the diamond chain. First, $J_{ab}^{[1]}(R)$ are identical for the three sets $(B, B)$, $(B, C)$ and

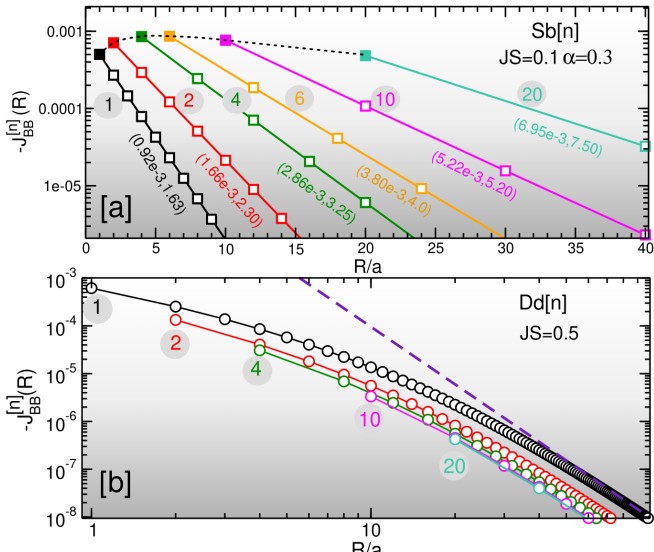

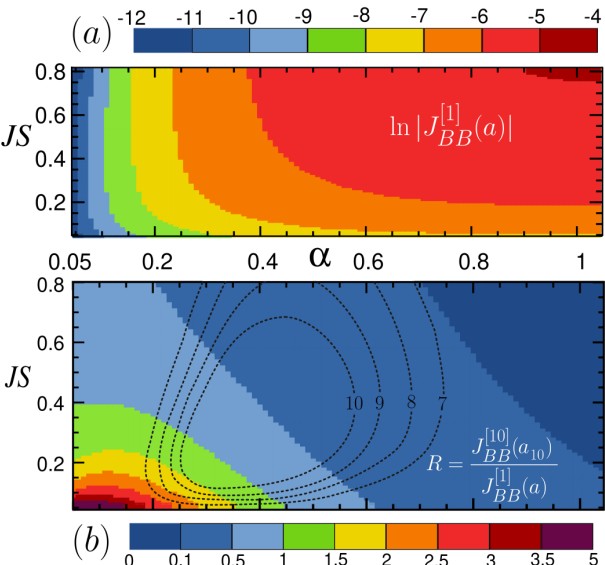

Figure 3. $-J_{BB}^{[n]}(R)$ as a function of $R$ in (a) Sb[n] and (b) Dd[n] chains. $JS$, $\alpha$ and $n$ are depicted. In (a), for each $n$, data are fitted by $A_n e^{-R/\xi_n}$, $(A_n, \xi_n)$ are indicated in the figure. The dashed line in (a) connects the n-n coupling in Sb[n] chains. The dashed (purple) line in (b) is an analytic expression (discussed in the text).

Figure 4. Two dimensional color plots of (a) $\ln|J_{BB}^{[1]}(a)|$ and (b) the ratio $r = \frac{J_{BB}^{[10]}(a_{10})}{J_{BB}^{[1]}(a)}$ in the $(\alpha, JS)$-plane. In panel (b), black dashed lines represent contour lines for $J_{BB}^{[10]}(a_{10})$ expressed in Kelvin, and assuming $t = 1\ eV$.

$(C, C)$ as could be anticipated since $B$ and $C$-orbitals are equivalent. The only difference concerns the $(B, C)$ coupling at $R = 0$, its large value is controlled by the FB-FB contribution. In contrast to the stub chain, in the limit of large $R$ the couplings cannot be fitted by a function that decays exponentially. It is found that $J_{ab}^{[1]}(R) = -C_1/R^4$ when $R \gg \sqrt{32}\frac{t}{JS}a$ where, as demonstrated in the SM the constant $C_1 = \frac{3}{2\pi}\frac{t^2}{|JS|}$. This analytical expression is depicted in Fig. 3(b). At large distances, the coupling is controlled by the contribution that originates from the two dispersive bands in the $\uparrow$ and $\downarrow$ sectors that touch at the zone boundary, all other contributions being exponentially small.

We propose to discuss the impact of the removal of $B$-orbitals. Naively, a sharp suppression of the couplings would be expected as the distance between $(B, B)$ pairs increases. However, as will be seen, under certain conditions, this scenario breaks down in the presence of flat bands. In contrast, the couplings can significantly increase which is counter-intuitive and defies all common sense. Fig. 3(a) depicts $J_{BB}^{[n]}(R)$ as a function of $R$ in Sb[n] chains with $n$ varying from 1 to 20. Here, $JS = 0.1$ and $\alpha = 0.3$ are chosen. The effect of tuning $JS$ and $\alpha$ will be discussed later on. First, as $n$ varies, $J_{BB}^{[n]}(R)$ stay ferromagnetic since the bipartite character of the lattice is preserved. In all cases, the couplings decay exponentially. Astonishingly, as $n$ increases one finds that the amplitude of the nearest neighbour coupling $|J^{[n]}(R = a_n)|$ in Sb[n] chains increase as well. It reaches a maximum for $n \approx 5$ beyond which it decays very slowly. It can

be seen that even in the Sb[10] chain $|J^{[10]}(a_{10})|$ is still larger than the nearest-neighbour coupling in Sb[1]. Furthermore, it should be noted that this coupling is about three orders of magnitude larger than that in the reference chain (Sb[1]) at the same distance. Thus, the removal of $B$-atoms leads to a giant amplification of the nearest-neighbour coupling which is in contradiction with the naive expectation. In addition, for any value of $n$ the data can be fitted perfectly by $J_{BB}^{[n]}(R) = -A_n e^{-R/\xi_n}$ where both $A_n$ and $\xi_n$ grow rapidly as $n$ increases. Unfortunately, in contrast to the case $n = 1$, the analytical derivation of the couplings is much more difficult in dilute chains since the number of terms in Eq. (3) scales as $n^2$ which makes it arduous to identify the dominant contributions.

What about the dilute diamond chain? Data are shown in Fig. 3(b) for $JS = 0.5$. Contrary to the dilute stub chain and as intuition suggests, we find that the coupling between nearest-neighbors pairs in the Dd[n] is smaller than that of the reference chain for the same distance. In addition, in all cases $(n \geq 1)$, at large distances, one observes that $J_{BB}^{[n]}(R) = C_n/R^4$ where $C_n$ decreases as $n$ increases. A question arises: How can we understand the difference of behavior between the dilute stub and diamond chains? In Sb[n] chains, the CLS spreads as the distance between nearest neigbours $B$-orbitals grows (see Fig. 1) which implies the increase of the QM [26]. In contrast, as discussed previously, in Dd[n] chains, the CLS are localized inside the unit cell (no overlap between CLS), which leads to a vanishing QM.

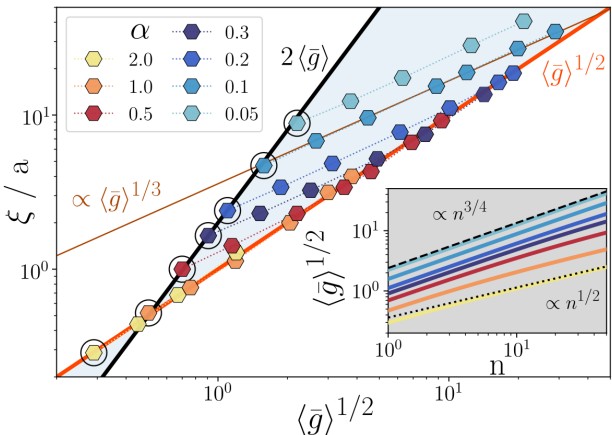

Figure 5. $\xi$ in Sb[n] chains as a function of $\langle \bar{g} \rangle^{1/2}$ where the dimensionless $\langle \bar{g} \rangle = \langle g \rangle / a^2$, $\langle g \rangle$ being the average of the QM. $JS = 0.1$, values of $\alpha$ are depicted. Red (resp. black) line is $\langle \bar{g} \rangle^{1/2}$ (resp. $2\langle \bar{g} \rangle$). Brown line is a guide to the eye needed for the discussion. Circled symbols correspond to $n = 1$. Inset shows $\langle \bar{g} \rangle^{1/2}$ as a function of $n$ for the same values of $\alpha$.

From now on, we focus on Sb[n] chains exclusively. We propose to estimate the impact of $JS$ and $\alpha$ on the nearest-neighbour coupling in the dilute chains. we consider the case of the Sb[10] chain and realize a two dimensional color plot of the ratio $r = \frac{J_{BB}^{[10]}(a_{10})}{J_{BB}^{[1]}(a)}$ in the $(\alpha, JS)$-plane. The result is depicted in Fig. 4(b). First, we observe that the largest values of $r$ are obtained for both $\alpha$ and $JS$ as small as possible. However, as can be seen in Fig. 4(a), this region is restricted to small values of $|J_{BB}^{[1]}(a)|$. Nevertheless, as highlighted by the contour lines in Fig. 4(b), large values of $J_{BB}^{[10]}(a_{10})$ can be achieved for parameters covering a substantial region in the $(\alpha, JS)$-plane. Remark that, for such large distances, $J_{BB}^{[1]}(10a)$ is expected to be several orders of magnitude smaller than $J_{BB}^{[10]}(a_{10})$. From Fig. 3(a), assuming $t = 1$ $eV$, $\alpha = 0.3$ and $JS = 0.1$, one would find $J_{BB}^{[1]}(a_{10}) = 10^{-2}$ $K$ which is 3 orders of magnitude smaller than $J_{BB}^{[10]}(a_{10})$.

In the last section, our purpose is to connect the decaying length $\xi$ as plotted in Fig. 3 to the average value of the QM given by $\langle g \rangle = \frac{1}{2\pi} \int_{-\pi}^{+\pi} g(k) dk$ where we recall that $g(k) = \langle \partial_k \Psi_k^{FB} | \partial_k \Psi_k^{FB} \rangle - |\langle \partial_k \Psi_k^{FB} | \Psi_k^{FB} \rangle|^2$, $|\Psi_k^{FB}\rangle$ being the FB eigenstate at $JS = 0$ [15]. Figure 5 depicts $\xi$ as a function of $\langle g \rangle^{1/2}$ (relevant characteristic length-scale) in half-filled Sb[n] chains for $JS = 0.1$ and different values of $\alpha$. We emphasize that the QM is calculated numerically using the same procedure as described in Ref. [27]. However, for $n = 1$, one can analytically show that $\langle g \rangle = \frac{a^2}{2\alpha\sqrt{\alpha^2+4}}$ [26]. First, for $n = 1$, we observe two distinct regimes: $\xi = 2\langle g \rangle / a$ for $\alpha \le 1$ and $\xi = \sqrt{\langle g \rangle}$ when $\alpha \ge 1$. This cross-over can be understood by considering the structure of $|CLS\rangle_{Sb[1]}$. The

weight is dominant on $B$-orbitals when $\alpha \ll 1$, and on $C$-orbitals when $\alpha \gg 1$. Hence, two successive CLSs overlap significantly only when $\alpha \ll 1$ which implies that the range of $J_{BB}(R)$ increases in this case and reduces in the other. In the SM [31] we have shown analytically the connection between $\xi$ and $\langle g \rangle$. We note that such a type of behaviour has been observed in the context of FB superconductivity in the Sb[1], where it has been shown that the superfluid weight scales linearly or quadratically with $\langle g \rangle$ depending on $\alpha$ [32]. We now consider dilute chains ($n > 1$). Again, we observe two distinct regimes depending on whether $\alpha$ is smaller or larger than 1. If $\alpha \ge 1$, we find that $\xi = \langle g \rangle^{1/2}$ while the situation differs drastically when $\alpha < 1$. In this regime, $\xi$ scales as $\langle g \rangle^{1/3}$ as long as $n$ is smaller than an $\alpha-$dependent integer $n_c(\alpha)$, while $\xi = \langle g \rangle^{1/2}$ when $n > n_c(\alpha)$. Note that $n_c(\alpha)$ increases rapidly as $\alpha$ reduces. More precisely, our data show that $n_c(\alpha) = 5, 10$ and $80$ for $\alpha = 0.5, 0.3$ and $0.1$ which suggests the scaling $n_c(\alpha)\alpha^2 \simeq 1$ when $\alpha \le 1$ and $n_c(\alpha) = 1$ for $\alpha \ge 1$. We attempt a simple explanation for the $\langle g \rangle^{1/3}$ scaling observed numerically assuming that the FB-FB contribution dominates. According to the expression of $|CLS\rangle_{Sb[n]}$, the weight on the $B$-orbitals varies in $1/\sqrt{n\alpha^2 + 2}$, suggesting that the analytical calculation performed in the reference case ($n = 1$) could be generalized by replacing $\alpha$ by $\sqrt{n}\alpha$ which would lead to $\xi \approx a_n / 2\sqrt{n}\alpha = \sqrt{n}a/2\alpha$. The inset of Fig. 5 shows that $\langle g \rangle^{1/2} \sim n^{3/4}$ for small values of $\alpha$, which nicely explains the singular $\langle g \rangle^{1/3}$ scaling. However, it is difficult to provide a simple explanation to $\xi \sim \langle g \rangle^{1/2}$ observed when $n > n_c(\alpha)$. The number of contributions to the couplings increases in $n^2$, thus the identification of the dominant terms is non-trivial. The previous explanation relies essentially on the fact that the FB-FB term dominates which should not be the case if $n$ becomes too large.

In conclusion, we have investigated the impact of flat bands on magnetic couplings in one-dimensional spin chains. We have highlighted a counter-intuitive and unexpected phenomenon: couplings between localized spins can increase dramatically as the distance between them increases. We were able to show that the quantum metric $\langle g \rangle$ characterizing flat band eigenstates is at the origin of these surprising results. It turns out that the QM controls the range of the couplings, highlighting complex scaling of the decay length with $\langle g \rangle$. We believe that our work could be of particular interest to research devoted to qubit entanglement and quantum communication, and hope that it will motivate experimental studies possibly using metallic organic frameworks to synthesize this type of magnetic structure in order to confirm the spectacular effects highlighted in our study.

We would like to thank E. Lorenzo, T. Deutsch and J. Colbois for the interesting discussions and their relevant remarks.

[32] M. Thumin and G. Bouzerar, Phys. Rev. B **107**, 214508 (2023).

* georges.bouzerar@neel.cnrs.fr

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

# Supplemental Material for "Generating long-distance magnetic couplings with flat bands" by G. Bouzerar and M. Thumin

The aim of this supplemental material is to provide analytical expressions of the $(B, B)$ couplings at large distances in the half-filled standard stub chain and diamond lattice. Here, the temperature is set to $0\ K$. Our calculations of the couplings focus only on non-dilute chains ($n = 1$). The case of larger values of $n$ is analytically much more difficult to handle because the number of contributions to the couplings increases in $n^2$ and the identification of the dominant terms becomes non-trivial.

## SPECTRUM AT $JS \neq 0$

Before discussing the nature of the couplings, we propose to briefly examine the spectrum at non-zero $JS$. The Hamiltonian is given by Eq.(1) in the main text. In Fig. S1 the dispersions in Sb[n] and Dd[n] chains ($n = 1, 4$) are depicted for $JS = 1$. The total number of bands (in each spin sector) being $2n + 1$, a part of the dispersions is depicted. For Sb[1] and Sb[4] $\alpha$ is set to 1 (see main text). In both systems, the FBs are located at $E_{FB} = \pm\frac{JS}{2}$. In each spin sector, the FB is isolated in Sb[n] chains, while in Dd[n] chains a band touching occurs at $k = 0$ or $\pi$ depending on $n$ even or odd. In addition, in Sb[n] chains at $E = 0$ a gap is found between the last filled band and the first empty one in the opposite spin sector. In contrast, these two bands touch quadratically at $k = 0$ ($n$ even) or $k = \pi$ ($n$ odd) in Dd[n] chains. These features characterize the Dd[n] and Sb[n] chains in general.

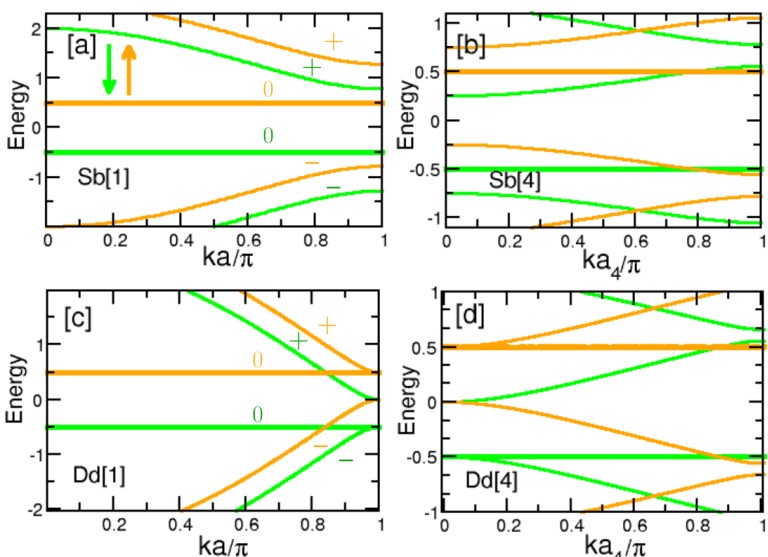

Figure S1. Dispersions ($\uparrow$ and $\downarrow$ sectors) in Sb[n] and Dd[n] chains ($n = 1$ and 4). $JS = 1$ and the flat bands (highlighted in thick lines) are located at $E_{FB} = \pm\frac{JS}{2}$. In Sb[1] and Sb[4], $\alpha = 1$. In [b] and [d], the size of the cell is $a_4 = 4a$.

## COUPLINGS IN THE STANDARD STUB CHAIN

Here, we consider $JS \geq 0$, the conclusions are the same if $JS \leq 0$, one just has to invert the $\uparrow$ and $\downarrow$ sectors. In the basis $(c^\dagger_{Ak\sigma}, c^\dagger_{Bk\sigma}, c^\dagger_{Ck\sigma})$, the $3 \times 3$ spin resolved Hamiltonian reads,

$$\hat{H}_\sigma = \begin{bmatrix} 0 & -\alpha t & -2t\cos(ka/2) \\ -\alpha t & z_\sigma\frac{JS}{2} & 0 \\ -2t\cos(ka/2) & 0 & z_\sigma\frac{JS}{2} \end{bmatrix}, \tag{S1}$$

where $z_\sigma = \pm 1$ for $\sigma = \uparrow$ or $\downarrow$.

The three eigenvalues are,

$$E_k^{0\sigma} = z_\sigma \frac{JS}{2}, \tag{S2}$$

$$E_k^{\pm\sigma} = \frac{1}{2}\left(z_\sigma \frac{JS}{2} \pm \sqrt{\Delta_\sigma^k}\right), \tag{S3}$$

with $\Delta_\sigma^k = (\frac{J^2 S^2}{4} + 4\alpha^2 t^2 + 16 t^2 \cos^2(ka/2))$.

These bands are depicted in Fig.(S1) [a] for $\alpha = 1$ and $JS = 1$. The associated eigenvectors are,

$$|\psi_k^{0\sigma}\rangle = \frac{1}{(\alpha^2 + 4\cos^2 ka/2)^{1/2}} \begin{bmatrix} 0 \\ -2\cos(ka/2) \\ \alpha \end{bmatrix}, \tag{S4}$$

and those of the dispersive bands,

$$|\psi_k^{\pm\sigma}\rangle = \frac{1}{\sqrt{N_\sigma^\pm}} \begin{bmatrix} \frac{1}{2t}\left(-z_\sigma \frac{JS}{2} \pm \sqrt{\Delta_\sigma^k}\right) \\ -\alpha \\ -2\cos(ka/2) \end{bmatrix}, \tag{S5}$$

where $N_\sigma^\pm = 8\cos^2(ka/2) + 2\alpha^2 + \frac{J^2 S^2}{8t^2} \mp \frac{1}{4}z_\sigma \frac{JS\sqrt{\Delta_\sigma^k}}{t^2}$.

We recall that the couplings are given by Eq. (3) and Eq. (4) in the main text. More precisely, the (B,B) coupling can be decomposed into different inter-band contributions,

$$J_{BB}(R) = \sum_{pq} I_{BB}^{pq}(R), \tag{S6}$$

$$I_{BB}^{pq}(R) = \frac{(JS)^2}{2N_c^2} \sum_{k,k'} e^{i(k-k')R} A_{kB,k'B}^{pq} \frac{f(E_k^{p\uparrow}) - f(E_{k'}^{q\downarrow})}{E_k^{p\uparrow} - E_{k'}^{q\downarrow}}, \tag{S7}$$

where $N_c$ is the number of unit cells, $R$ is the distance between the $B$ atoms and $p$ (resp. $q$) labels the band in the $\uparrow$ (resp. $\downarrow$) sector, and $A_{kB,k'B}^{pq} = \langle B_{k,\uparrow}|\psi_k^{p\uparrow}\rangle\langle\psi_k^{p\uparrow}|B_{k,\uparrow}\rangle\langle B_{k',\downarrow}|\psi_{k'}^{q\downarrow}\rangle\langle\psi_{k'}^{q\downarrow}|B_{k',\downarrow}\rangle$ with $|B_{k,\sigma}\rangle = \hat{c}_{Bk\sigma}^\dagger|vac\rangle$. Finally $f(E)$ is the Fermi-Dirac occupation.

We find that the gap $\delta$ between the last filled band and the lowest empty one (in the opposite spin sector) is $\delta = JS$ if $\alpha \geq \frac{JS}{\sqrt{2}t}$ otherwise $\delta = \sqrt{\frac{J^2 S2}{4} + 4\alpha^2 t^2}$ for $\alpha \leq \frac{JS}{\sqrt{2}t}$. Hence, when $\alpha \gg JS/t$ the couplings are expected to be dominated by the FB-FB term $I_{BB}^{00}(R)$ while they reduce to $I_{BB}^{-+}(R)$ when $\alpha \ll JS/t$. In these two limiting regimes, couplings can be calculated analytically, which is the aim of the next two subsections.

### (B,B) couplings for $\alpha \gg JS/t$

As noted above, in the regime $\alpha \gg JS/t$, the dominant contribution is the FB-FB term $I_{BB}^{00}(R)$. The other contributions being much smaller, we can safely write,

$$J_{BB}(R) = -\frac{JS}{2}|F_{00}(R)|^2, \tag{S8}$$

where we introduce,

$$F_{00}(R) = \frac{1}{N} \sum_k \frac{4\cos^2(ka/2)}{4\cos^2(ka/2) + \alpha^2} e^{ikR}. \tag{S9}$$

Because $E_k^{0\uparrow} - E_k^{0\downarrow} = JS$ and the eigenstates $|\psi_k^{0\sigma}\rangle$ are $JS$-independent, $J_{BB}(R)$ is linear in $JS$. This contrasts with the usual quadratic dependence of the couplings in the RKKY regime [S1].

$F_{00}(R)$ can be rewritten ($R/a \geq 1$),

$$F_{00}(R) = -\frac{\alpha^2}{2\pi} \int_{-\pi}^{+\pi} dk e^{ikR} \frac{1}{\alpha^2 + 2 + e^{ik} + e^{-ik}}. \tag{S10}$$

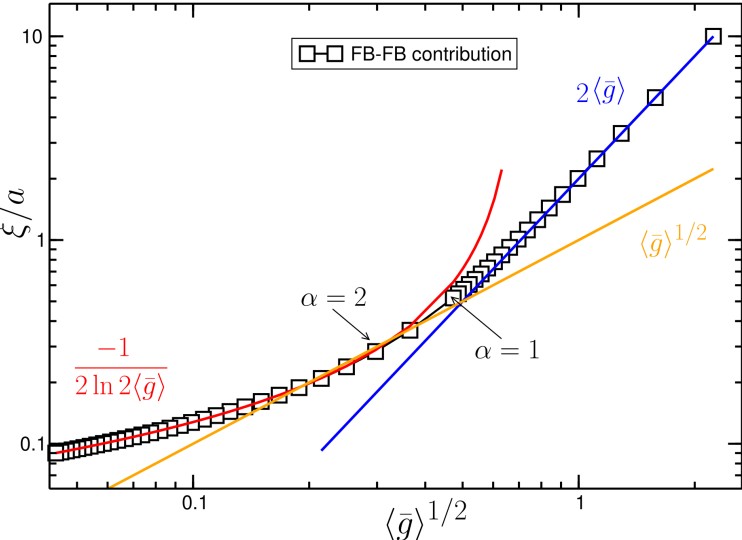

Figure S2. Analytical expression of $\xi$, obtained from the FB-FB contribution to $J_{BB}(R)$ (Eq.(S11)), as a function of $\langle \bar{g} \rangle = \langle g \rangle / a^2$, $\langle g \rangle$ being the average of the quantum metric. The red and blue lines represent the asymptotic behaviors in the limit of small and large $\langle g \rangle$. The orange line as displayed in Fig. 6 of the main text is plotted to facilitate the discussion.

This integral can be calculated exactly for any value of $R$ using a standard residue calculation which leads to,

$$F_{00}(R) = \frac{\alpha}{\sqrt{\alpha^2 + 4}} \exp\left(R \ln |z_-|\right), \tag{S11}$$

where the relevant pole is $z_+ = -1 - \frac{\alpha^2}{2} + \frac{\alpha}{2}(\alpha^2 + 4)^{1/2}$.
First, when $\alpha$ is small enough $(\alpha \le 1)$, Eq.(S11) can be simplified and gives,

$$J_{BB}(R) = -\frac{JS}{8}\alpha^2 \exp\left(-R/\xi\right), \tag{S12}$$

where the decaying length is,

$$\xi/a = -\frac{1}{2}\ln(|z_+|) \approx \frac{1}{2\alpha}. \tag{S13}$$

We have verified that these analytical expressions (Eq. (S12) and Eq. (S13)) are accurate by a direct comparison with our complete numerical calculations.
In the opposite limit $(\alpha \gg 1)$, using the expression of $z_+$, one now finds,

$$J_{BB}(R) = -\frac{JS}{2} \exp\left(-R/\xi\right), \tag{S14}$$

with,

$$\xi/a = \frac{1}{2\ln(\alpha^2)}. \tag{S15}$$

We briefly discuss the connection between $\xi = -\frac{1}{2}\ln(|z_+|)$ and the average value of the quantum metric $\langle g \rangle$ (see main text). We recall that $\sqrt{\langle g \rangle}$ can be interpreted as a measure of the typical spread of flat-band eigenstates [S2, S3]. Figure S2 depicts $\xi$ as a function of $\sqrt{\langle g \rangle}$, where one can show that $\langle g \rangle = \frac{a^2}{2\alpha\sqrt{\alpha^2+4}}$ [S4]. We clearly distinguish two different regimes: (i) $\xi = 2\langle g \rangle/a$ for $\alpha \ll 1$, and (ii) $\xi = -\frac{1}{2\ln(2\langle g \rangle/a^2)}$ for $\alpha \gg 1$. These (analytical) data coincide almost perfectly with those plotted in Fig. 6 of the main text, where $\xi$ has been extracted from the complete numerical calculation of $J_{BB}$. Finally, notice that the data plotted in Fig.6 could suggest that $\xi = \langle g \rangle^{1/2}$ for $\alpha > 1$ (orange line in Fig. S3). Numerically, we could not consider $\alpha$ values greater than 2 because of the tiny amplitude of $\xi$ in this regime $(\xi \sim 0.1a)$. As clearly shown analytically, the correct scaling is $\xi = -\frac{1}{2\ln(2\langle g \rangle/a^2)}$.

## (B,B) couplings for $\alpha \ll JS/t$

In this subsection, we now consider the opposite limit, $\alpha \ll JS/t$. In this case, the dominant contribution is expected to be $I_{BB}^{-+}(R)$ which is given by,

$$J_{BB}(R) = I_{BB}^{-+}(R) = \frac{(JS)^2}{2} F_{-+}(R), \tag{S16}$$

with,

$$F_{-+}(R) = \frac{1}{N^2} \sum_{k,k'} e^{i(k-k')R} \frac{A_{kB,k'B}^{-\uparrow,+\downarrow}}{E_k^{-\uparrow} - E_{k'}^{+\downarrow}}, \tag{S17}$$

where $A_{kB,k'B}^{-\uparrow,+\downarrow} = |\langle B_{k\uparrow}|\psi_k^{-\uparrow}\rangle|^2 |\langle B_{k'\downarrow}|\psi_{k'}^{+\downarrow}\rangle|^2$.

The dominant weight to the double sum originates from the vicinity of the zone boundary ($ka \approx k'a \approx \pm\pi$). Thus Eq. (S17) can be simplified and becomes,

$$F_{-+}(R) = -\frac{C}{N^2} \sum_{q,q'} e^{i(q-q')R} \frac{1}{1 + \frac{1}{2\alpha^2}(q^2 + q'2)}, \tag{S18}$$

where the prefactor $C = \frac{4\alpha^2 t^2}{(JS)^3}$.

Eq. (S18) can be rewritten,

$$F_{-+}(R) = -\frac{C\alpha^2}{2\pi^2} \int_{-\infty}^{\infty} \int_{-\infty}^{\infty} e^{iu} e^{-iv} \frac{dudv}{2\alpha^2 R^2 + u^2 + v^2}. \tag{S19}$$

First, we perform an integration with respect to the variable $v$ (residue calculation), and get,

$$F_{-+}(R) = -\frac{C\alpha^2}{\pi} \Re\left[\int_0^\infty e^{iu} du \frac{e^{-\sqrt{u^2+2\alpha^2 R^2}}}{\sqrt{u^2 + 2\alpha^2 R^2}}\right]. \tag{S20}$$

Using the fact that [S5],

$$\int_0^\infty dx x^{2n} \frac{e^{-\sqrt{x^2+b^2}}}{\sqrt{x^2+b^2}} = (2n-1)!!(b)^n K_n(b), \tag{S21}$$

where $K_n$ are the modified Bessel functions of second kind.

After replacing in Eq. (S20) $e^{iu}$ by $\sum_n \frac{1}{n!}(iu)^n$ one finds,

$$F_{-+}(R) = -\frac{C\alpha^2}{\pi} \sum_n \frac{(-1)^n}{n!} \left(\frac{\sqrt{2}\alpha R}{2}\right)^n K_n(\sqrt{2}\alpha R). \tag{S22}$$

Using the fact that for any $n$, $K_n(b) = \sqrt{\frac{\pi}{2b}} e^{-b}$ ($b > 0$) in the limit of large argument, we finally end up with,

$$J_{BB}(R) = \frac{2\alpha^{7/2}}{\sqrt{2\sqrt{2}\pi}} \frac{t^2}{JS} \frac{e^{-\frac{R}{\xi}}}{\sqrt{R}}, \tag{S23}$$

where in this regime ($\alpha t \ll JS$) we find,

$$\xi = \frac{\sqrt{2}}{3\alpha}. \tag{S24}$$

We have checked that this expression coincides very well with the data plotted in Fig.3 of the main text for $JS = 1$ and $\alpha = 0.1$ and $R$ being large enough.

## $(B, B)$ COUPLING IN THE DIAMOND CHAIN

In this section, our purpose is to show that $J_{BB}(R) = -C_1/R^4$ when $R$ is large enough.

In the basis $(c^\dagger_{Ak\sigma}, c^\dagger_{Bk\sigma}, c^\dagger_{Ck\sigma})$, the $3 \times 3$ Hamiltonian of the standard diamond chain (Dd[1]) reads,

$$
\hat{H}_\sigma = \begin{bmatrix} 0 & \epsilon_0(k) & \epsilon_0(k) \\ \epsilon_0(k) & z_\sigma \frac{JS}{2} & 0 \\ \epsilon_0(k) & 0 & z_\sigma \frac{JS}{2} \end{bmatrix},
\tag{S25}
$$

where we define $\epsilon_0(k) = -2t\cos(ka/2)$ and $z_\sigma = \pm 1$ for $\sigma = \uparrow$ or $\downarrow$.

The eigenvalues are straightforwardly calculated,

$$
E_k^{0\sigma} = z_\sigma \frac{JS}{2},
\tag{S26}
$$

$$
E_k^{\pm\sigma} = \frac{1}{2}\left(z_\sigma \frac{JS}{2} \pm \sqrt{\Delta_\sigma^k}\right),
\tag{S27}
$$

with $\Delta_\sigma^k = (\frac{J^2 S^2}{4} + 8\epsilon_0^2(k))$.

These bands are depicted in Fig.(S1) [b] for $JS = 1$. The corresponding eigenstates are given by,

$$
|\psi_k^{0\sigma}\rangle = \frac{1}{\sqrt{2}}\begin{bmatrix} 0 \\ 1 \\ -1 \end{bmatrix},
\tag{S28}
$$

and,

$$
|\psi_k^{\pm\sigma}\rangle = \frac{1}{\sqrt{N_\sigma^\pm}}\begin{bmatrix} \frac{1}{2}(z_\sigma \frac{JS}{2} \mp \sqrt{\Delta_\sigma^k}) \\ -\epsilon_0(k) \\ -\epsilon_0(k) \end{bmatrix},
\tag{S29}
$$

where $N_\sigma^\pm = 4\epsilon_0^2(k) + \frac{J^2 S^2}{8} \mp \frac{1}{4}z_\sigma JS\sqrt{\Delta_\sigma^k}$.

These expressions imply that the dispersive bands $(\uparrow, -)$ (filled band) and $(\downarrow, +)$ (empty band) touch at $ka = \pm\pi$. Thus, we anticipate that $I_{BB}^{-+}$ (see Eq. S6) (i) is the main contribution to the couplings (at large distances) and (ii) it decays as a power law. The other contributions are negligible since they involve pairs of bands $(p, q)$ separated by finite gaps, and thus are exponentially suppressed.

Using the same notation as that of the previous section, we write

$$
J_{BB}(R) = I_{BB}^{-+}(R) = \frac{(JS)^2}{2}F_{-+}(R),
\tag{S30}
$$

where,

$$
F_{-+}(R) = \frac{1}{N^2}\sum_{k,k'} e^{i(k-k')R}\frac{\epsilon_0^2(k)}{N_\uparrow^-(k)}\frac{\epsilon_0^2(k')}{N_\downarrow^+(k')}\frac{1}{E_\uparrow^-(k) - E_\downarrow^+(k')}.
\tag{S31}
$$

The main contribution to the double sum originates from the vicinity of the band touching (quadratic). Hence, we substitute $k = q \pm \pi$ and $k' = q' \pm \pi$ which leads to,

$$
F_{-+}(R) = -\frac{t^2}{\pi^2(JS)^3}\int_{-\pi}^{\pi}\int_{-\pi}^{\pi} dq\,dq'\,e^{i(q-q')R}\frac{q^2 q'^2}{q^2 + q'^2}.
\tag{S32}
$$

First, we perform the integral with respect to $q'$ which gives,

$$
F_{-+}(R) = \frac{t^2}{\pi(JS)^3}\int_{-\pi}^{\pi} dq\,e^{iqR}q^2|q|e^{-|q|R}.
\tag{S33}
$$

Focusing on $R/a \gg 1$ we get,

$$
F_{-+}(R) = \frac{2t^2}{\pi(JS)^3}\frac{1}{R^4}\Re\left[\int_0^\infty du\,e^{iu}u^3 e^{-u}\right].
\tag{S34}
$$

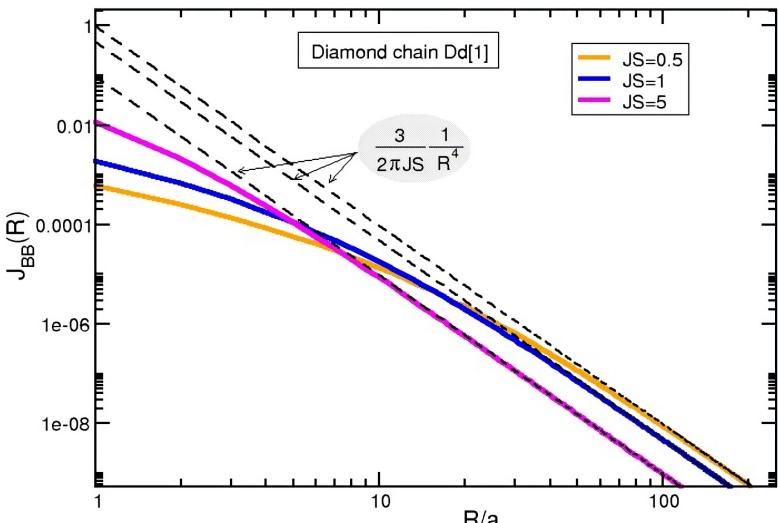

Figure S3. $J_{BB}(R)$ (numerical values) in the half-filled diamond chain as a function of $R$ for different values of $JS$. The dashed lines correspond to the analytical expression as given in Eq.(S35).

This clearly shows the power law decay, the (B,B) coupling scales as $1/R^4$.
Using the fact $\int_0^\infty x^n e^{-\alpha x} dx = n! \alpha^{-n-1}$ when $\Re(\alpha) > 0$, we then end up with the final expression,

$$J_{BB}(R) = -\frac{3}{2\pi} \frac{t^2}{JS} \frac{1}{R^4}. \tag{S35}$$

This analytical expression is valid only when $R \gg \sqrt{32} t/JS$ which is the condition to write Eq. (S32). In addition, the $1/JS$ scaling does not depend on the value of $JS$. Consequently, in the weak coupling regime ($JS \ll t$) the RKKY expression, which predicts a $(JS)^2$ dependence, fails. The comparison between the complete numerical calculation of $J_{BB}(R)$ and Eq.(S35) is displayed in Fig.S2. As illustrated, they coincide when $R$ is sufficiently large.
It is also interesting to note that because the flat-band eigenstates are $k$-independent (Eq. (S28)), $I_{BB}^{pq}(R)$ is zero for $R/a \geq 1$ when at least one of the bands ($p$ or $q$) is a flat-band. With the exception of the contribution to $J_{BC}(R=0)$, flat bands do not contribute and can be ignored.

* georges.bouzerar@neel.cnrs.fr
[S1] Bouzerar, G. Flat band induced room-temperature ferromagnetism in two-dimensional systems. Phys. Rev. B. **107**, 184441 (2023,5), https://link.aps.org/doi/10.1103/PhysRevB.107.184441
[S2] Marzari, N. & Vanderbilt, D. Maximally localized generalized Wannier functions for composite energy bands. Phys. Rev. B. **56**, 12847-12865 (1997,11), https://link.aps.org/doi/10.1103/PhysRevB.56.12847
[S3] Marzari, N., Mostofi, A., Yates, J., Souza, I. & Vanderbilt, D. Maximally localized Wannier functions: Theory and applications. Rev. Mod. Phys.. **84**, 1419-1475 (2012,10), https://link.aps.org/doi/10.1103/RevModPhys.84.1419
[S4] Bouzerar, G. Giant boost of the quantum metric in disordered one-dimensional flat-band systems. Phys. Rev. B. **106**, 125125 (2022,9), https://link.aps.org/doi/10.1103/PhysRevB.106.125125
[S5] G. Watson, A Treatise on the Theory of Bessel Functions. Cambridge Mathematical Library, Cambridge University Press, 1995.