# Peer review of "Generating long-distance magnetic couplings with flat bands"

_SciPost Physics_

## Round 1 · Referee Report · Anonymous (Referee 1) · 2025-4-20

Strengths

Originality and novelty: Clearly formulated and innovative idea – connecting the quantum metric of flat-band systems with the spatial range of long-range effective magnetic couplings.

Consistency and correctness: The results align with known physical principles, supported by analytical derivations and numerical simulations.

Relevance: The study addresses long-standing open questions in quantum communication and magnetic material design. The suggestion to use MOF architectures for realization adds practical value.

Detailed modeling: Two different 1D geometries (stub and diamond chains) are studied, with a clear demonstration of how geometry affects the quantum metric and coupling range.

Weaknesses

Idealized modeling: The approach assumes a perfectly flat band exactly at the Fermi level and zero temperature. In real materials, even slight dispersion, randomness or doping may suppress the predicted effects.

Missing references: The manuscript omits some recent relevant studies that analyze similar 1D geometries and flat-band systems. Among these, one could for instance mention: R. R. Montenegro-Filho and M. D. Coutinho-Filho, Phys. Rev. B 90, 115123 (2014); V. M. Martinez Alvarez and M. D. Coutinho-Filho, Journal of Physics: Condensed Matter 31, 195603 (2019); R. R. Montenegro-Filho, D. R. B. Silva, D. Cogollo, M. D. Coutinho-Filho, Phys. Rev. B 111, 054416 (2025).

Classical spins only: The model assumes classical spins and neglects quantum fluctuations, which could be significant in systems with S = 1/2 (e.g., possible entanglement effects).

Ferromagnetic coupling only: The predicted long-range couplings are ferromagnetic. It remains unclear how these could be used to generate singlet-based entanglement needed for quantum information processing.

Report

The manuscript provides an original and well-founded theoretical study on a mechanism to generate long-range effective magnetic couplings in spin chains through flat electronic bands. The authors demonstrate that the quantum metric of the flat-band eigenstates directly determines the length scale over which magnetic exchange between distant spins remains significant. In certain configurations, they observe a counterintuitive enhancement of these couplings at long distances. The work is motivated by the challenge of achieving long-range quantum entanglement, relevant for quantum communication, and proposes that such effects could be realized in real materials such as MOF (metal-organic framework) structures.

In conclusion, this is a strong theoretical contribution to the field of flat-band magnetism and quantum spin interactions. The manuscript opens a new perspective and may have significant impact on the design of quantum materials and quantum communication networks. I recommend publication after addressing the points below.

Requested changes

  1. Include a discussion on the robustness of the effect with respect to flat-band dispersion, doping, randomness, and temperature. What is
  2. Cite and compare with recent works, who studied similar chains and effects using a different methodology. For instance, the systems similar to diamond chain were studied in R. R. Montenegro-Filho and M. D. Coutinho-Filho, Phys. Rev. B 90, 115123 (2014); V. M. Martinez Alvarez and M. D. Coutinho-Filho, Journal of Physics: Condensed Matter 31, 195603 (2019); R. R. Montenegro-Filho, D. R. B. Silva, D. Cogollo, M. D. Coutinho-Filho, Phys. Rev. B 111, 054416 (2025). The stub chain considered in this work seems to be also equivalent to the so-called branched chain.
  3. Comment on possible ways to utilize ferromagnetic couplings in quantum architectures – e.g., by dynamically switching the interaction or combining with auxiliary elements to generate entanglement.
  4. Provide at least rough estimates of key physical parameters needed for MOF-based realization – e.g., how perfectly flat must the band be, and what energy scales are expected.
  5. The notion of diamond chain is usually reserved for the case when there is interaction or hopping term between B and C sites. Consider changing notation from the diamond chain to AB2 chain as used previously by many authors. Similarly, the stub chain could be alternatively changed to a branched chain.
  6. How sensitive is the observed long-range coupling to flat-band dispersion and deviations from half-filling? Could small perturbations in realistic MOF systems significantly weaken the effect?
  7. What practical mechanism do the authors envision for using the predicted ferromagnetic interactions to establish quantum entanglement between distant qubits?
  8. Which effect is expected to result from quantum fluctuations if the classical spins would be changed to the quantum ones?

Recommendation

Ask for minor revision

---

## Round 1 · Referee Report · Yeongjun Kim (Referee 2) · 2025-5-2

Report

The authors investigate how the real-space range of the exchange interaction \(J\) between localized spins on flat-band lattices depends on (i) the average of quantum metric (QM) and (ii) the gap to the nearest dispersive bands. A striking outcome is that the \(B\!-\!B\) coupling does \textit{not} diminish—even though the distance between neighbouring \(B\) orbitals increases with the dilution parameter \(n\).
This counter-intuitive behaviour is traced to the structure of the flat-band eigenstates, whose growing quantum metric enhances both the coupling amplitude and its decay length.
The analysis is technically sound and novel enough, and I believe the work merits publication after the very minor points listed below are addressed.

Requested changes

  • Definition of $R=a_n$. The notation $R=a_n$ appears in Fig. 3(a) but is never defined.
    Please add a one-line explanation.

  • Cross-references to the Supplementary Material.
    Whenever a central statement relies on derivations in the SM, please cite the exact SM section or equation.
    For instance, the sentence “At large distances, the coupling is controlled by the contribution that originates from the two dispersive bands …” explains the algebraic $1/R^{4}$ tail in the diamond chain, but the supporting calculation is only in SM.
    Adding an explicit pointer there (and in similar places) will greatly help readers.

Attachment

Recommendation

Publish (surpasses expectations and criteria for this Journal; among top 10%)

---

## Editorial Decision

resubmitted